# Protective Effects of Ambroxol on Aβ and α-Synuclein-Induced Neurotoxicity Through Glucocerebrosidase Activation in HT-22 Hippocampal Neuronal Cells

**DOI:** 10.3390/ijms252212114

**Published:** 2024-11-12

**Authors:** Sheng-Chieh Lin, Ching-Chi Chang, Sing-Hua Tsou, Pai-Yi Chiu, Ju-Fang Cheng, Hui-Chih Hung, Wei-Jen Chen, Ying-Jui Ho, Chih-Li Lin

**Affiliations:** 1School of Medicine, Chung Shan Medical University, Taichung 402, Taiwan; phoenix33343@gmail.com (S.-C.L.); fmaj7@seed.net.tw (C.-C.C.); 2Department of Orthopedics, Chung Shan Medical University Hospital, Taichung 402, Taiwan; 3Department of Psychiatry, Chung Shan Medical University Hospital, Taichung 402, Taiwan; 4Department of Medical Research, Chung Shan Medical University Hospital, Taichung 402, Taiwan; zinminid@gmail.com; 5Department of Neurology, Show Chwan Memorial Hospital, Changhua 500, Taiwan; paiyibox@gmail.com; 6Division of Pediatric Neurology, Changhua Christian Hospital, Changhua 500, Taiwan; babygladys@gmail.com; 7Department of Life Sciences and Institute of Genomics and Bioinformatics, National Chung Hsing University, Taichung 402, Taiwan; hchung@dragon.nchu.edu.tw; 8Department of Biomedical Sciences, Chung Shan Medical University, Taichung 402, Taiwan; cwj519@csmu.edu.tw; 9Department of Psychology, Chung Shan Medical University, Taichung 402, Taiwan; 10Institute of Medicine, Chung Shan Medical University, Taichung 402, Taiwan

**Keywords:** αSyn (αSyn), Ambroxol (AMBX), amyloid β (Aβ), autophagy, glucocerebrosidase

## Abstract

Dementia with Lewy bodies (DLB) is a progressive neurodegenerative disorder marked by the accumulation of α-synuclein (αSyn), often co-existing with amyloid β (Aβ) pathology. Current treatments are largely symptomatic, highlighting a critical need for disease-modifying therapies. Evidence suggests that αSyn aggregates contribute to neuronal death in DLB, particularly when exacerbated by Aβ. Given the role of autophagy in clearing misfolded proteins, exploring agents that promote this pathway is essential for developing effective treatments. Ambroxol (AMBX), a mucolytic drug, has demonstrated potential in activating glucocerebrosidase (GCase), an enzyme that enhances lysosomal function and facilitates the autophagic clearance of toxic protein aggregates, including αSyn. This study aims to evaluate AMBX’s neuroprotective effects in a cellular model of DLB, with the goal of identifying new therapeutic agents that target the underlying pathology of DLB. In this study, HT-22 hippocampal neuronal cells were exposed to αSyn and Aβ, followed by AMBX treatment. Our results showed that AMBX significantly improved cell viability and reduced apoptosis in cells co-treated with αSyn and Aβ. Additionally, AMBX restored GCase activity, promoted autophagy, and reduced oxidative stress, which in turn mitigated αSyn aggregation and phosphorylation. These findings suggest that by activating GCase and enhancing autophagy, AMBX may help alleviate DLB-associated neurodegeneration. This study underscores the potential of AMBX as a therapeutic agent for DLB and supports further investigation in animal models and clinical trials to validate its efficacy in neurodegenerative disease contexts.

## 1. Introduction

Dementia with Lewy bodies (DLB) is a progressive neurodegenerative disorder, recognized as the second most common form of dementia after Alzheimer’s disease (AD) [1]. It is characterized by the accumulation of α-synuclein (αSyn) protein in the brain, forming Lewy bodies that disrupt neurotransmission, especially within acetylcholine and dopamine pathways. This disruption leads to a range of cognitive, motor, sleep, and behavioral symptoms [2]. DLB shares clinical features with both AD and Parkinson’s disease (PD), such as fluctuating cognition, visual hallucinations, parkinsonism, and rapid eye movement (REM) sleep behavior disorder, which complicate accurate diagnosis and differentiation [3]. Pathologically, Lewy bodies are frequently found in the cerebral cortex, midbrain, and brainstem and are observed in individuals with PD, about 50% of AD patients, and other neurodegenerative cases [4]. Although the etiology of DLB remains unclear, most cases appear sporadic, with only rare familial occurrences reported. Currently, there is no cure or treatment to slow the progression of DLB, though symptom management can improve quality of life. Research continues to seek potential biomarkers and disease-modifying therapies, but no definitive treatment has been identified thus far.

The primary component of Lewy bodies, αSyn, is a 140-amino acid protein integral to synaptic function. However, when it misfolds and aggregates, αSyn leads to neuronal damage in DLB [5]. While most cases of DLB are sporadic, rare familial forms have been linked to mutations in genes such as *SNCA* and *GBA* [6]. The accumulation of misfolded αSyn in the brain leads to neuronal death and drives DLB progression. Genetic factors influencing AD, such as the *APOE* gene, are also involved in DLB, with APOE being the most significant genetic risk factor for sporadic cases of DLB, highlighting the genetic overlap between these disorders [7]. Additionally, amyloid β (Aβ), a protein central to AD, has been shown to exacerbate αSyn aggregation, suggesting a co-pathological mechanism in which both proteins drive disease progression [8]. Numerous studies support this association, indicating that Aβ plays a critical role in advancing DLB pathology. For instance, Lloyd et al. showed that Aβ deposits in the brain are associated with localized αSyn aggregation in transgenic mouse models, reinforcing the notion that Aβ exacerbates αSyn pathology [9]. Similarly, Colom-Cadena et al. found significant overlap between Aβ and αSyn deposits in the cortical regions of DLB patients, further emphasizing the co-pathological relationship in neurodegeneration [10]. Our previous studies have also confirmed that Aβ facilitates αSyn aggregation, creating a disease model that mirrors the pathology of DLB [11,12]. This co-aggregation disrupts normal neuronal function and interferes with protective mechanisms such as oxidative stress and autophagy, both essential for maintaining cellular health. The impairment of these pathways increases αSyn-induced cytotoxicity, thereby worsening neurodegeneration in DLB. Consequently, Aβ can be recognized as a critical factor in modeling DLB pathology, highlighting the shared pathological mechanisms between DLB and AD [13].

Effective therapeutic strategies for DLB should focus on the clearance of aggregated αSyn from neurons. The ubiquitin–proteasome system (UPS) and autophagy are the primary intracellular pathways for clearing misfolded proteins [14]. However, the UPS is limited in its ability to degrade aggregated αSyn, making autophagy—a process for degrading and recycling cellular debris—a more promising mechanism for clearing these toxic aggregates [15]. Among known autophagy activators, Ambroxol (AMBX), a common mucolytic agent, has shown potential as a neuroprotective agent in synucleinopathies [16]. Beyond its conventional use, AMBX acts as a pharmacological chaperone by enhancing lysosomal activity and increasing glucocerebrosidase (GCase) levels, an enzyme crucial for degrading misfolded proteins, including αSyn [17]. In studies on synucleinopathies, particularly PD, AMBX has been shown to reduce αSyn aggregation in neurons, improving neuronal health and reducing disease severity. This effect is thought to be mediated by its activation of autophagy, promoting the natural degradation of toxic protein aggregates. However, its effects on DLB have not yet been explored. This gap in research forms the basis of the present study, which aims to evaluate the potential of AMBX as a therapeutic agent for DLB by investigating its role in autophagy activation and αSyn clearance in neuronal cells.

## 2. Results

### 2.1. Synergistic Neurotoxicity of αSyn Overexpression and Aβ in HT-22 Cells

Figure 1A illustrates the experimental framework using the GeneSwitch system, which enables controlled and time-specific αSyn overexpression through mifepristone induction. In this setup, cells were transfected with either a control or the inducible vector system. The pGene vector contains the αSyn gene, while the pSwitch vector enables regulated gene overexpression in response to mifepristone stimulation. Upon mifepristone treatment, the pSwitch vector activates the pGene vector, resulting in αSyn overexpression. In contrast, the control cells lack an inserted gene in the pGene vector and therefore do not undergo any gene overexpression, serving as a control for comparison with the αSyn overexpression model. The validation of this system is shown in Figure 1B, where Western blot analysis confirmed that 48 h of mifepristone treatment resulted in a dose-dependent increase in αSyn expression, peaking at 1 μM. This concentration was subsequently used in all experiments to investigate the interaction between αSyn and Aβ. Bright-field microscopy (Figure 1C) showed that αSyn overexpression alone caused no significant morphological damage to HT-22 cells. In contrast, treatment with a low concentration of Aβ (2.5 μM) induced mild cellular damage, consistent with its known neurotoxic effects. However, when αSyn overexpression was combined with Aβ treatment, a significant increase in cellular damage was observed, suggesting that αSyn sensitizes neurons to Aβ-induced toxicity. To quantify the impact on cell viability, Figure 1D shows results from an AO/PI viability assay. In this analysis, green bars represent the percentage of live cells, while red bars represent the percentage of apoptotic cells. The data indicated that while αSyn overexpression alone did not significantly reduce cell viability, Aβ treatment moderately increased apoptosis and co-exposure to αSyn and Aβ markedly heightened cell death. These findings suggest that the combined presence of αSyn and Aβ exacerbates neurotoxicity, supporting the hypothesis that their interaction plays a crucial role in advancing neurodegeneration, as seen in DLB.

### 2.2. Ambroxol Attenuates Aβ and α-Synuclein-Induced Apoptosis and Aggregation

The MTT viability assay (Figure 2A) demonstrated that co-treatment with Aβ and αSyn significantly reduced HT-22 cell survival to 51%, whereas the control group showed normal viability. Treatment with AMBX improved cell viability in a dose-dependent manner, with a maximum effect at 20 μM, increasing viability to approximately 72%. Western blot analysis (Figure 2B) further revealed that Aβ and αSyn co-treatment elevated apoptotic markers, including cleaved caspase-3 and PARP, indicating increased apoptosis. AMBX treatment significantly reduced these apoptotic markers, suggesting that AMBX preserves cell viability by inhibiting apoptosis. Analysis of αSyn solubility (Figure 2C) showed that Aβ treatment significantly increased insoluble αSyn levels, promoting aggregation. AMBX, however, reduced the accumulation of insoluble αSyn, suggesting its role in preventing pathological aggregation. Importantly, soluble αSyn levels remained consistent across treatment conditions, indicating AMBX’s specific effect on the aggregation-prone form. Immunofluorescence imaging and high-content analysis (HCA) (Figure 2D) showed that αSyn overexpression alone slightly increased small aggregates (<1 μm) without affecting larger aggregates (≥1 μm). In contrast, Aβ treatment substantially increased large, toxic aggregates, which were effectively reduced by AMBX. These findings suggest that AMBX mitigates αSyn aggregation induced by Aβ, potentially offering neuroprotective benefits in conditions such as DLB, where the co-aggregation of Aβ and αSyn is central to disease progression.

### 2.3. Ambroxol Restores Autophagic Function and GCase Activity in Aβ and αSyn-Treated HT-22 Cells

AO staining results (Figure 3A) showed alterations in acidic vesicular organelles (AVOs) under various treatment conditions. In control and mifepristone-treated cells, AVOs were prominent, indicated by deep orange-red fluorescence. However, co-treatment with Aβ and αSyn significantly reduced AVOs, suggesting impaired autophagic vesicle formation. AMBX treatment restored the number of AVOs, indicating that AMBX reversed the inhibitory effects of Aβ and αSyn on autophagy. Western blot analysis was conducted to examine key autophagic proteins, including LC3-II, Atg7, Atg12-Atg5, and p62 (Figure 3B). Co-treatment with Aβ and αSyn significantly decreased the expression of autophagic degradation markers LC3-II, Atg7, and Atg12-Atg5, while increasing p62 levels, an indicator of impaired autophagic flow. These results suggest that autophagy was inhibited, likely contributing to toxic aggregate accumulation. Conversely, AMBX treatment restored these autophagic markers, indicating that AMBX alleviated autophagic dysfunction and enhanced aggregate clearance. Additionally, GCase activity was measured (Figure 3C), showing that Aβ and αSyn co-treatment reduced GCase activity, impairing lysosomal degradation. AMBX co-treatment partially restored GCase activity, promoting lysosomal function and the clearance of misfolded proteins through the autophagy–lysosome pathway. Western blot analysis of GCase protein levels (Figure 3D) revealed that Aβ and αSyn treatment slightly reduced GCase expression, consistent with reduced GCase activity. AMBX modestly increased GCase levels, underscoring that its primary effect lies in enhancing GCase function rather than protein expression, further supporting its role in restoring autophagic efficiency.

### 2.4. GCase Activation Is Essential for Ambroxol’s Protective Effects Against Aβ and αSyn-Induced Cytotoxicity in HT-22 Cells

To validate the role of GCase activation in AMBX’s neuroprotective effects, the GCase-specific inhibitor conduritol B epoxide (CBE) was employed in subsequent experiments. The MTT viability assay results (Figure 4A) showed that while AMBX mitigated the toxicity of combined Aβ and αSyn treatment, its protective effect was significantly reduced upon CBE addition, suggesting that AMBX’s benefits depend on GCase activation. Immunofluorescence staining (Figure 4B) indicated that AMBX reduced αSyn aggregation, which was less effective when CBE was present. This supports the notion that functional GCase is critical for limiting αSyn aggregation. ROS levels were assessed using DCFH-DA staining (Figure 4C), showing that AMBX treatment lowered ROS-positive cells, thereby reducing oxidative stress. However, CBE co-treatment negated this effect, increasing ROS-positive cells, further indicating that AMBX’s antioxidative benefits hinge on GCase activation. Lastly, Western blot analysis examined αSyn phosphorylation at Ser^129^ (Figure 4D), a modification linked to Lewy bodies and oxidative stress. AMBX reduced Ser^129^ phosphorylation, potentially alleviating αSyn pathology associated with Lewy body formation [18]. In contrast, CBE inclusion raised Ser^129^ phosphorylation levels, suggesting that GCase inhibition may exacerbate oxidative stress and promote pathogenic αSyn modifications. These results collectively underscore the essential role of GCase activation in enabling AMBX to confer neuroprotection, effectively reducing oxidative stress, αSyn aggregation, and phosphorylation—key factors implicated in DLB neurodegeneration.

## 3. Discussion

Our study highlights that αSyn overexpression alone does not induce cytotoxicity in neuronal cells, emphasizing that while αSyn aggregation is central to DLB pathology, the presence of Aβ is crucial for significant neurotoxicity. This finding aligns with the existing literature suggesting that αSyn requires additional factors, such as Aβ, to fully express its pathogenic potential [19]. Consistent with prior research, we observed that αSyn overexpression in HT-22 cells did not provoke marked neurotoxicity or aggregation on its own. However, when combined with low concentrations of Aβ, substantial cytotoxicity and aggregation occurred, underscoring Aβ’s role as a pathological amplifier. This result supports clinical observations that AD and DLB frequently coexist, indicating shared or complementary pathologies influenced by Aβ. The synergy between Aβ and αSyn in our model, where Aβ significantly exacerbated αSyn’s toxic effects, aligns with evidence suggesting that both AD and DLB share co-pathological mechanisms. Recent studies propose that Aβ may facilitate αSyn’s pathological transformation, potentially through impaired autophagy and increased oxidative stress [8], reinforcing the need for DLB therapies that target both proteins.

Further, our findings demonstrate that AMBX significantly reduces Aβ-induced αSyn aggregation, likely through enhanced autophagic activity. Autophagy is increasingly recognized as a critical pathway for the degradation of misfolded proteins like αSyn, particularly given the limitations of the ubiquitin–proteasome system in handling such aggregates due to their complex structures [14]. This aligns with a broader scientific consensus that autophagy provides a more effective route for αSyn clearance. AMBX’s capacity to enhance autophagy positions it as a promising candidate for alleviating neurodegenerative burdens in DLB and other conditions characterized by αSyn aggregation. Importantly, our data suggest that AMBX’s effect on GCase activity is central to its ability to promote autophagic flux and facilitate αSyn clearance. GCase is essential for lysosomal function, and its deficiencies are linked to synucleinopathies like PD and DLB [20]. By activating GCase, AMBX appears to support lysosomal degradation of αSyn, mitigating aggregation and associated toxicity. Research on Gaucher disease and PD further indicates that AMBX-mediated increases in GCase activity improve lysosomal function and reduce αSyn levels [21]. Although AMBX influences both autophagy and GCase activity, GCase activation appears to be more critical, as it drives autophagic flux, thus supporting effective clearance of αSyn aggregates [22].

Understanding the role of oxidative stress and its relationship to autophagic dysfunction and GCase activity is essential for addressing DLB pathology and identifying therapeutic targets [23]. Oxidative stress damages neuronal cells directly and disrupts autophagy, a crucial process for degrading misfolded proteins like αSyn. This disruption perpetuates oxidative damage, creating a harmful cycle that worsens neurodegeneration in DLB [24]. GCase is vital for lysosomal function, enabling the breakdown of glucocerebrosides and misfolded proteins. In synucleinopathies like Gaucher disease, reduced GCase activity impairs lysosomes, limiting autophagy and allowing αSyn aggregates to accumulate, which in turn contributes to oxidative stress. This interplay creates a self-reinforcing loop, linking oxidative stress, autophagic dysfunction, and αSyn aggregation as central factors in PD pathology [25]. AMBX offers a potential intervention by acting as a molecular chaperone that activates GCase [26]. By stabilizing GCase, AMBX helps restore lysosomal function and supports autophagy. The clearance of aggregated αSyn facilitated by AMBX can alleviate oxidative stress as it improves cellular homeostasis and reduces cytotoxicity [27]. The clearance of aggregated αSyn facilitated by AMBX can alleviate oxidative stress, improving cellular homeostasis and reducing cytotoxicity. This is particularly significant because oxidative stress is known to increase αSyn’s aggregation tendency, thereby contributing to DLB pathology [28]. Moreover, oxidative stress can weaken lysosomal membranes, further impairing autophagy. By enhancing GCase function, AMBX may strengthen lysosomes, making them more resilient to oxidative damage and thereby enabling a more effective autophagic response. [29]. Through its dual role in reducing oxidative stress and enhancing autophagy, AMBX presents a comprehensive neuroprotective approach.

Our research builds on the understanding of AMBX’s neuroprotective potential, which has been previously documented in PD. As a mucolytic agent, AMBX has been widely used for years, with a well-established safety profile and low risk of toxicity even at higher doses, making it a promising candidate for repurposing in new therapeutic applications [30]. Notably, Ambroxol is able to cross the blood–brain barrier (BBB), a critical factor for its potential use in neurological conditions. Clinical trials have demonstrated Ambroxol’s efficacy in patients with neuronopathic Gaucher disease, where it reduced neuroinflammation and improved neurological outcomes, further supporting its potential for treating neurodegenerative diseases involving central nervous system pathology [31]. By extending its known benefits to DLB, we highlight AMBX as a viable therapeutic candidate for targeting αSyn pathology, a key factor in both PD and DLB. Considering the limited treatment options available for DLB, our findings strongly suggest further exploration of AMBX’s therapeutic potential in this context. While our in vitro results provide robust evidence of AMBX’s protective effects, there are inherent limitations to this model. Future studies should validate these results in animal models, followed by clinical trials to establish AMBX’s efficacy in humans. A deeper understanding of the molecular pathways through which AMBX activates GCase and autophagy will also be essential for optimizing its therapeutic use. In summary, our study illustrates how AMBX effectively reduces Aβ-induced αSyn aggregation and neurotoxicity through mechanisms involving GCase activation and autophagy enhancement. These findings highlight the therapeutic promise of AMBX in addressing the overlapping pathologies of AD and DLB by targeting both αSyn aggregation and Aβ toxicity. Further research on AMBX’s dual effects on αSyn and Aβ co-pathology could offer new avenues for treating DLB and related neurodegenerative diseases.

## 4. Materials and Methods

### 4.1. Materials

The chemicals used in the study included Ambroxol (AMBX, purity ≥ 98%), MTT, mifepristone (MFP), acridine orange (AO), propidium iodide (PI), and conduritol B epoxide (CBE), all sourced from Sigma-Aldrich (München, Germany). Primary antibodies for αSyn (αSyn), phosphorylated Ser^129^-αSyn, caspase-3, PARP, and p62 were purchased from GeneTex (Irvine, CA, USA), while antibodies specific to β-actin and LC3 were acquired from Novus Biologicals (Littleton, CO, USA). Additional antibodies for GCase, Atg7, and Atg12-Atg5 were obtained from Cell Signaling Technology (Danvers, MA, USA). The αSyn (*SNCA*, GeneID: 6622) coding sequence was provided by transOMIC (Huntsville, AL, USA), and Aβ1-42 peptides were synthesized by LifeTein (Somerset, NJ, USA). The Aβ solution was prepared as per previously established protocols. All reagents were pre-dissolved in either phosphate-buffered saline (PBS) or dimethyl sulfoxide (DMSO), based on their solubility properties, and stored at temperatures below −20 °C until needed for experimentation.

### 4.2. Cell Culture and Transfection

The immortalized mouse hippocampal cell line HT-22 cells were obtained from Merck (Darmstadt, Germany). Cells were cultured in Dulbecco’s Modified Eagle Medium (DMEM) supplemented with 10% fetal bovine serum (FBS) and 1% penicillin-streptomycin (Thermo Fisher Scientific, Waltham, MA, USA) at 37 °C in a humidified atmosphere with 5% CO_2_. To overexpress αSyn, cells were transfected with the GeneSwitch™ System (Invitrogen, Carlsbad, CA, USA), establishing an inducible HT-22 cell line. Cells were seeded at a density of 1 × 105 cells per well in 6-well plates in DMEM supplemented with 10% FBS and incubated for 24 h before transfection. The coding sequence of human αSyn was inserted into the inducible expression plasmid pGene. For stable transfection, cells were first transfected with the regulatory plasmid pSwitch, which constitutively expresses a mifepristone (MFP)-responsive GAL4 fusion protein (GAL4-DBD/hPR-LBD/p65-AD), using Lipofectamine 2000 (Thermo Fisher Scientific, Waltham, MA, USA). Cells were then selected with hygromycin over four weeks to ensure stable integration. Next, cells were co-transfected with pGene-αSyn, followed by selection with zeocin for another four weeks, resulting in a doubly stable HT-22 cell line expressing both regulatory and inducible constructs. To induce αSyn overexpression, stable cells were treated with 1 μM MFP for 48 h prior to further experimental treatments as specified in each experimental setup.

### 4.3. Western Blot Analysis

Cells were lysed in a buffer containing 50 mM Tris-HCl (pH 8.0), 5 mM EDTA, 150 mM NaCl, 0.5% Nonidet P-40, 0.5 mM DTT, 1 mM PMSF, 0.15 U/mL aprotinin, 5 μg/mL leupeptin, 1 μg/mL pepstatin, and 1 mM NaF. The lysate was first centrifuged at 1000× *g* for 10 min at 4 °C to collect the soluble αSyn in the supernatant. The remaining pellet, containing non-soluble αSyn, was resuspended in 2% SDS buffer and further centrifuged at 12,000× *g* for 30 min at 4 °C. Both supernatants were then prepared for immunoblotting. For Western blot analysis, equal amounts of total protein (50 μg) were separated by SDS-PAGE and transferred to polyvinylidene difluoride (PVDF) membranes (Millipore, Bedford, MA, USA). Membranes were blocked and incubated with primary antibodies at a 1:1000 dilution in 0.1% Tween-20, followed by incubation with horseradish peroxidase-conjugated secondary antibodies at a 1:5000 dilution. The immunoreactive bands were detected using an enhanced chemiluminescence detection system (Millipore, Bedford, MA, USA), and protein levels were quantified by densitometric analysis using Quantity One software version 4.6.2 (Bio-Rad, Hercules, CA, USA). Protein expression was normalized to β-actin, and relative expression levels were calculated by setting the control sample level to 100% for comparison across samples.

### 4.4. Determination of Cell Viability Using Acridine Orange (AO)-Propidium Iodide (PI) Staining

Cell viability was assessed using the AO-PI staining kit (Logos Biosystems, Annandale, VA, USA). Acridine orange (AO, 10 µg/mL) permeates all cells and produces green fluorescence, while propidium iodide (PI, 5 µg/mL) can only enter dead cells, resulting in red fluorescence by binding to nucleic acids and enhancing fluorescence 20–30 fold. After treatment under specified conditions, cells were trypsinized, suspended, and stained with 2 µL of AO-PI solution mixed with 18 µL of the cell sample. This mixture was analyzed immediately using the LUNA-FL Dual Fluorescence Cell Counter (Logos Biosystems, Annandale, VA, USA) according to the manufacturer’s protocol. Cell viability and apoptosis were quantified by averaging cell counts, and values were expressed as the percentage of dead cells relative to the total cell count.

### 4.5. MTT Cell Viability Assay

To assess cell viability, cells were seeded in a 24-well plate. After treatment, MTT reagent (5 mg/mL) was added to each well, and cells were incubated at 37 °C for 4 h to form purple formazan crystals in viable cells. The crystals were dissolved in DMSO, and absorbance was measured at 570 nm using a Jasco V-700 spectrophotometer (JASCO, Tokyo, Japan). Cell viability was calculated as a percentage relative to the control group, which was set at 100%.

### 4.6. Immunocytochemistry and AO Staining

For immunocytochemistry, cells were seeded in 24-well black imaging plates at a density of 1 × 106 cells per well and allowed to adhere. Following treatment, cells were fixed with 4% paraformaldehyde at 4 °C, with 1 µg/mL Hoechst 33342 in PBS for overnight incubation. After blocking with 1% BSA, cells were incubated with primary antibodies targeting αSyn, followed by Rhodamine-conjugated secondary antibodies. Fluorescence images were captured using the ImageXpress Micro Confocal High-Content imaging system (Molecular Devices, Sunnyvale, CA, USA). The quantification of αSyn aggregates was performed through automated analysis with MetaXpress software version 6.7.0.211 (Molecular Devices), which randomly selected 10 fields per well for rhodamine-positive particle counting and size measurement. To assess autophagic vesicles, cells were stained with AO and observed under a fluorescence microscope. Acidic vesicular organelles (AVOs) were identified by red fluorescence.

### 4.7. GCase Activity Assay

GCase activity was measured using a GCase Activity Assay Kit (Abcam, Cambridge, MA, USA). Cell lysates were prepared in the provided assay buffer, and GCase activity was determined by measuring fluorescence at 360 nm excitation and 445 nm emission using the SpectraMax iD5 microplate reader (Molecular Devices, Sunnyvale, CA, USA), following the kit protocol. The results were normalized to protein content to ensure accurate comparisons across samples.

### 4.8. Measurement of Reactive Oxygen Species (ROS)

ROS levels were assessed using the fluorescent dye DCFH-DA. Cells were incubated with DCFH-DA (10 µM) for 30 min at 37 °C, and fluorescence intensity was observed using the ImageXpress Micro Confocal High-Content imaging system (Molecular Devices, Sunnyvale, CA, USA). Fluorescence intensity was quantified using automated analysis with MetaXpress software (Molecular Devices, Sunnyvale, CA, USA), which randomly selected 10 fields per well to measure DCFH-DA staining on a per-cell basis. The results from all fields were averaged to determine overall oxidative stress levels.

### 4.9. Statistical Analysis

All data are presented as means ± standard deviation (SD). Statistical analyses were conducted using analysis of variance (ANOVA), followed by Tukey’s post-hoc test for multiple comparisons, using SPSS Statistics 25 (SPSS, Inc., Chicago, IL, USA). A *p*-value of less than 0.05 was considered statistically significant.

## Figures and Tables

**Figure 1 ijms-25-12114-f001:**
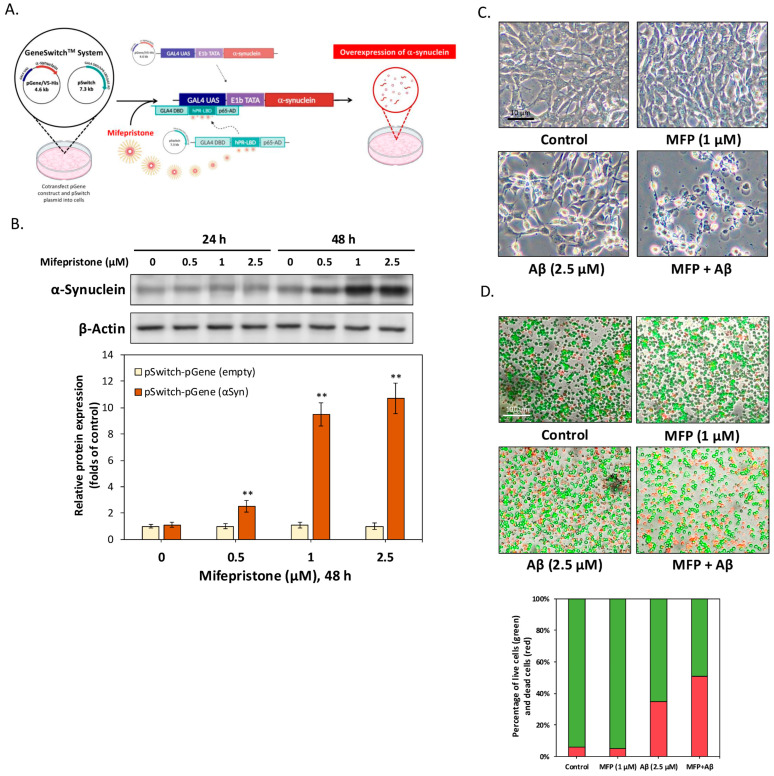
Enhanced cytotoxicity from combined αSyn overexpression and Aβ treatment in HT-22 cells. (**A**) Schematic of the Invitrogen GeneSwitch™ system used to induce αSyn overexpression in HT-22 cells via mifepristone. In the control cells, the pGene vector lacks an inserted gene at the expression site, so no protein is expressed, serving as the control. In contrast, the αSyn cells have the αSyn gene inserted in the pGene vector, leading to overexpression of αSyn protein upon mifepristone stimulation. (**B**) Western blot analysis showing a dose-dependent increase in αSyn expression after 48 h of mifepristone treatment, with peak levels at 1 μM. (**C**) Bright-field microscopy images indicating no significant morphological damage in cells overexpressing αSyn alone. However, mild damage was observed with Aβ (2.5 μM) treatment, and the combination of αSyn and Aβ led to exacerbated cellular damage. (**D**) AO/PI staining demonstrated that αSyn overexpression alone did not significantly affect cell viability. Green bars represent live cells and red bars represent apoptotic cells. Aβ treatment increased apoptosis, with the highest level observed when αSyn overexpression was combined with Aβ, indicating enhanced neurotoxicity. All data were collected from at least three independent experiments and presented as mean ± SD. An asterisk (*) indicates a significant difference compared with the control group (** *p* < 0.01).

**Figure 2 ijms-25-12114-f002:**
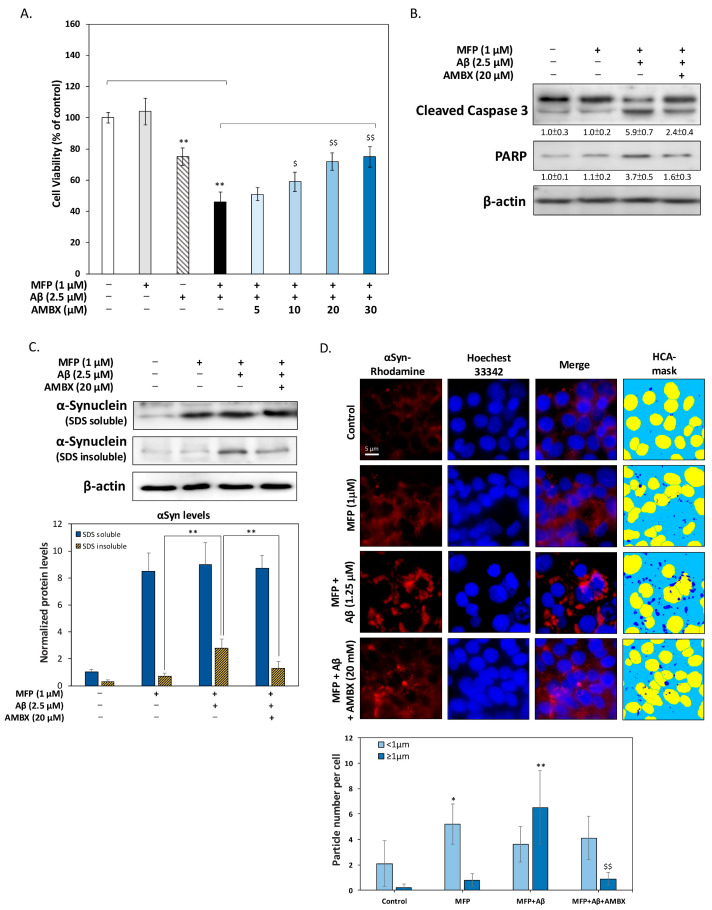
Ambroxol mitigates Aβ and αSyn-induced apoptosis and protein aggregation. (**A**) MTT cell viability assay results showed that co-treatment with Aβ and αSyn significantly reduced cell survival to 51%, compared to the control group. The addition of AMBX restored viability in a dose-dependent manner, increasing it to approximately 72%, with 20 μM being the most effective concentration. (**B**) Western blot analysis revealed that Aβ and αSyn co-treatment increased apoptotic markers (cleaved caspase-3 and PARP), indicating cell death through apoptosis. AMBX significantly reduced these markers, suggesting it inhibited apoptosis and protected neuronal cells. (**C**) Analysis of αSyn solubility showed that Aβ promoted the accumulation of insoluble αSyn, increasing its levels by 4–5 times compared to control conditions. AMBX reduced insoluble αSyn levels, but neither treatment affected soluble αSyn. (**D**) High-content analysis (HCA) of immunofluorescence images demonstrated that Aβ treatment increased large αSyn aggregates (≥1 μm), while AMBX significantly reduced their formation. In the HCA-mask images, the yellow color indicates the labeled nuclear regions of each cell, whereas the dark blue color represents the regions of aggregated αSyn. AMBX effectively targeted and reduced toxic αSyn aggregates induced by Aβ, mitigating neurotoxicity. All data were collected from at least three independent experiments and presented as mean ± SD. An asterisk (*) indicates a significant difference compared with the control group (* *p* < 0.05 and ** *p* < 0.01), and a dollar sign ($) indicates a significant difference compared with the Aβ and αSyn co-treatment group ($ *p* < 0.05 and $$ *p* < 0.01).

**Figure 3 ijms-25-12114-f003:**
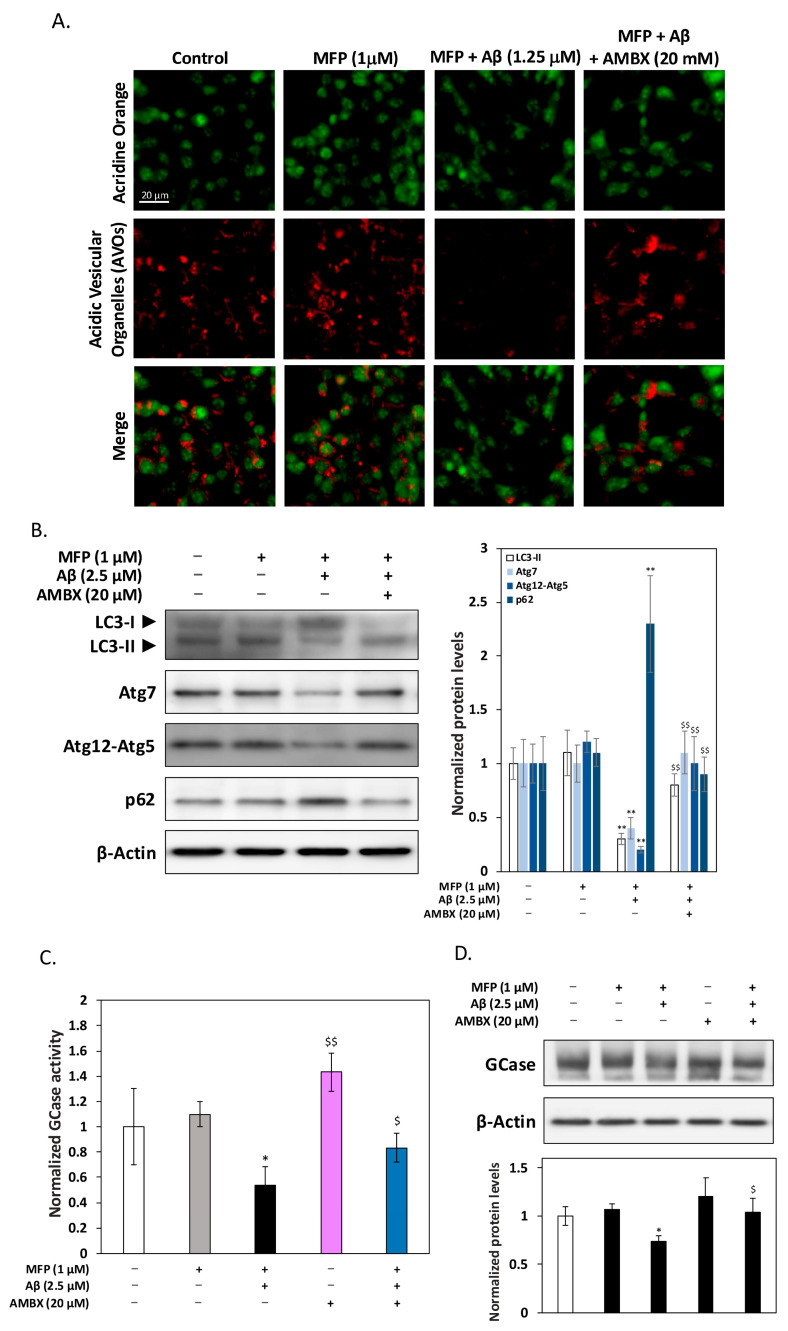
Ambroxol enhances autophagic function and GCase activity in HT-22 cells treated with Aβ and αSyn overexpression. (**A**) Acridine orange (AO) staining showed that Aβ and αSyn overexpression reduced acidic vesicular organelles (AVOs) in HT-22 cells, indicating suppressed autophagy. In the fluorescent images, the green color represents AO background staining, while the red color marks the AVOs. AMBX co-treatment restored AVOs, suggesting recovery of autophagic activity. (**B**) Western blot results indicated that Aβ and αSyn overexpression decreased autophagy-related proteins LC3-II, Atg7, and Atg12-Atg5, while increasing p62, a marker of autophagic impairment. AMBX reversed these changes, promoting autophagic protein clearance. (**C**) GCase activity assays demonstrated reduced GCase function with Aβ and αSyn overexpression, suggesting lysosomal impairment. AMBX partially restored GCase activity, supporting enhanced lysosomal degradation and protein clearance. (**D**) Western blotting showed a slight decrease in GCase expression due to Aβ and αSyn overexpression. AMBX modestly increased GCase levels, though its impact on activity was greater than on expression, emphasizing AMBX’s role in supporting autophagic function. All data were collected from at least three independent experiments and presented as mean ± SD. An asterisk (*) indicates a significant difference compared with the control group (* *p* < 0.05 and ** *p* < 0.01), and a dollar sign ($) indicates a significant difference compared with the Aβ and αSyn co-treatment group ($ *p* < 0.05 and $$ *p* < 0.01).

**Figure 4 ijms-25-12114-f004:**
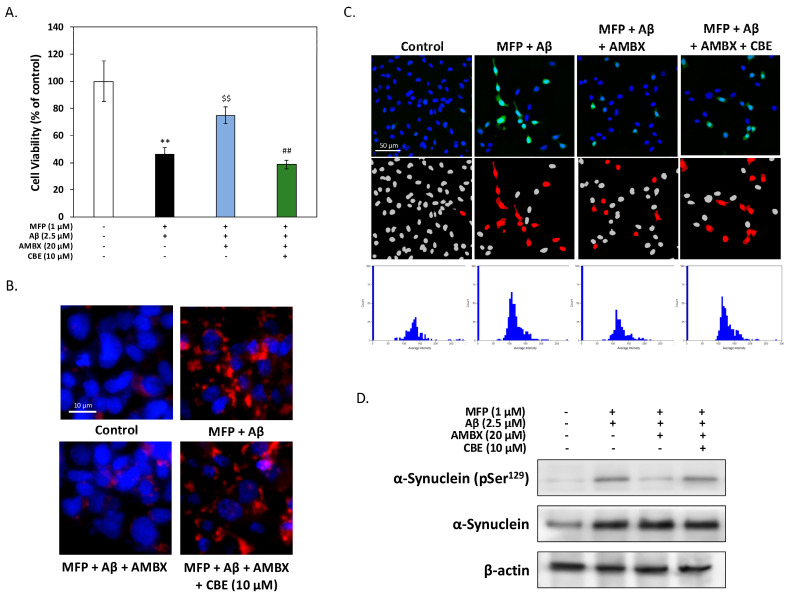
GCase activation is critical for AMBX’s protective effects against Aβ and αSyn cytotoxicity. (**A**) The MTT cell viability assay indicated that AMBX effectively protected HT-22 cells from Aβ and αSyn-induced toxicity. However, this protection was notably reduced with the addition of the GCase inhibitor, conduritol B epoxide (CBE), underscoring the necessity of GCase activity for AMBX’s protective effects. (**B**) Immunofluorescence staining showed that AMBX decreased αSyn aggregates in cells co-treated with Aβ and αSyn, but this reduction was less effective when CBE was present. These findings suggest that GCase activity is also essential for limiting αSyn aggregation. (**C**) DCFH-DA staining and high-content analysis (HCA) revealed that AMBX lowered intracellular ROS levels, mitigating oxidative stress from Aβ and αSyn exposure. This antioxidative effect was negated by CBE, indicating AMBX’s reliance on GCase to alleviate oxidative stress. (**D**) Western blot analysis demonstrated that AMBX reduced αSyn phosphorylation at Ser^129^, a modification linked to oxidative stress and Lewy body pathology. CBE reversed this reduction, suggesting that GCase inhibition may exacerbate oxidative stress and pathological phosphorylation. All data were collected from at least three independent experiments and presented as mean ± SD. An asterisk (*) indicates a significant difference compared with the control group (** *p* < 0.01), a dollar sign ($) indicates a significant difference compared with the Aβ and αSyn co-treatment group ($$ *p* < 0.01), and a pound sign (#) indicates a significant difference compared with the Aβ and αSyn co-treatment group (## *p* < 0.01).

## Data Availability

All the data are shown in the main manuscript.

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
