# Peer review of "Protective Effects of Ambroxol on Aβ and α-Synuclein-Induced Neurotoxicity Through Glucocerebrosidase Activation in HT-22 Hippocampal Neuronal Cells"

_ijms, 2024, doi:10.3390/ijms252212114_

Round 1

Reviewer 1 Report

Comments and Suggestions for Authors

This paper by Sheng-Chieh Lin et al demonstrated that Ambroxol significantly suppressed Aβ-induced α-synuclein aggregation via the activation of autophagy in TH 22 cells. The study appears to be of interest, whereas the experiments have several problems. In my opinion, therefore, this manuscript is not recommended for publication in its present form, but can accept as the paper after throughout revisions.

Major points

1)    Limitation

The protective effect of Ambroxol in the present study is admissible only in the HT22 cells nor another cells? I wonder that they can be extrapolated to the primary neurons or model mice? Please discuss these points in the revised manuscript.

2)    Discussion or Introduction

If you are discussing the validity of Ambroxol as a treatment for dementia, I feel strongly that it is necessary to discuss the BBB permeability and brain transport of Ambroxol. Please comment or modified this point in the revised manuscript.

3)    Data and statics

Dunnett's and Tukey's tests are both post-hoc multiple comparison tests used after an ANOVA, but they serve different purposes. Dunnett's test compares multiple treatment groups to a single control group, focusing on identifying significant differences between each treatment and the control. Tukey's test (or Tukey's HSD) compares all possible pairs of groups, assessing if any two groups differ significantly. I strongly recommend you that the post-tests should be selected Tukey's tests in this study.

4)    Fig. 1 legends.

The band data in Fig 1B is not fully explained in the present manuscript. Mock or pSwitch?

The graph in Fig 1D is also insufficiently explained. What are the differences in color?

Author Response

This paper by Sheng-Chieh Lin et al demonstrated that Ambroxol significantly suppressed Aβ-induced α-synuclein aggregation via the activation of autophagy in TH 22 cells. The study appears to be of interest, whereas the experiments have several problems. In my opinion, therefore, this manuscript is not recommended for publication in its present form, but can accept as the paper after throughout revisions.

Major points

Comment 1:

1)Limitation

The protective effect of Ambroxol in the present study is admissible only in the HT22 cells nor another cells? I wonder that they can be extrapolated to the primary neurons or model mice? Please discuss these points in the revised manuscript.

Response 1:

Thank you for this valuable comment. We agree that utilizing primary neurons would theoretically provide a more physiologically relevant model. However, in our study, we employed a genetically modified HT22 cell line designed for stable α-synuclein overexpression, allowing for consistent expression levels, which are crucial for our experimental design. Using primary neurons would necessitate transient transfection, which can result in variable α-synuclein expression, potentially compromising the stability of our model. Nevertheless, we acknowledge the importance of validating these findings in animal models to better approximate in vivo conditions. We will incorporate this limitation in the revised manuscript and emphasize our intent to address it in future research. We appreciate the reviewer’s suggestion on this point.

Comment 2:

2) Discussion or Introduction

If you are discussing the validity of Ambroxol as a treatment for dementia, I feel strongly that it is necessary to discuss the BBB permeability and brain transport of Ambroxol. Please comment or modified this point in the revised manuscript.

Response 2:

Thank you for highlighting the importance of discussing the blood-brain barrier (BBB) permeability of Ambroxol. Studies have shown that Ambroxol is able to cross the BBB, which supports its potential as a treatment for neurological conditions. For instance, it has been investigated in clinical trials as a pharmacological chaperone for neuronopathic Gaucher disease, where it demonstrated efficacy in reducing neuroinflammation and improving neurological outcomes​ (Hollander et. al., Mol Genet Meta 2024;143:108556).

We have included a discussion on Ambroxol’s BBB permeability and its evaluation in clinical trials in the revised manuscript to address this point. We appreciate the reviewer’s insightful comment, which has helped us improve the context and relevance of our work.

Comment 3:

3) Data and statics

Dunnett's and Tukey's tests are both post-hoc multiple comparison tests used after an ANOVA, but they serve different purposes. Dunnett's test compares multiple treatment groups to a single control group, focusing on identifying significant differences between each treatment and the control. Tukey's test (or Tukey's HSD) compares all possible pairs of groups, assessing if any two groups differ significantly. I strongly recommend you that the post-tests should be selected Tukey's tests in this study.

Response 3:

We sincerely appreciate the reviewer’s expert suggestion regarding the appropriate choice of post-hoc tests. We fully agree and have implemented Tukey’s test in place of Dunnett’s test in this revised version. Upon reanalysis, the significance levels of all comparisons remain unchanged. Additionally, we have updated the description in Section 4.9, Statistical Analysis, to reflect this modification. Thank you once again for this valuable correction.

Comment 4:

4) Fig. 1 legends.

The band data in Fig 1B is not fully explained in the present manuscript. Mock or pSwitch?

The graph in Fig 1D is also insufficiently explained. What are the differences in color?

Response 4: Thank you for your insightful comment. We have addressed this concern by providing a more detailed explanation of the control and experimental conditions in both the manuscript text and the Figure 1 legend. To avoid confusion, we have removed the term "Mock" and clarified that both control and αSyn cells were transfected with the pSwitch vector. Specifically, the control cells contain the pGene vector without an inserted gene, while the αSyn cells contain the α-synuclein gene in the pGene vector, allowing for α-synuclein overexpression upon mifepristone stimulation. This revision provides a clearer understanding of the experimental setup and control conditions. We appreciate the reviewer’s suggestion to improve clarity in our descriptions.

For Fig. 1D, we have updated both the main text and the figure legend to provide a detailed explanation of the green and red bars in the graph. Specifically, the green bars represent the percentage of live cells, while the red bars indicate the percentage of apoptotic cells. This clarification enhances the interpretability of the AO/PI staining results. We appreciate the reviewer’s suggestion to improve the clarity of our data presentation.

Reviewer 2 Report

Comments and Suggestions for Authors

the manuscript ijms-3286572 entitled Protective Effects of Ambroxol on Aβ and α-Synuclein-Induced Neurotoxicity Through GCase Activation in HT-22 Hippocampal Neuronal Cells by Sheng‐Chieh Lin, aims to evaluate the Ambroxol (AMBX) neuroprotective effects in a cellular model of DLB, addressing the motivation to identify new therapeutic agents that target the underlying pathology of DLB. In this study, HT-22 cells were exposed to α-synuclein and Aβ, followed by AMBX treatment. The results showed that AMBX significantly improved cell viability and reduced apoptosis in cells co-treated with α-synuclein and Aβ. Additionally, AMBX restored GCase activity, promoting autophagy and reducing oxidative stress, which in turn mitigated α-synuclein aggregation and phosphorylation. These findings suggest that by activating GCase and enhancing autophagy, AMBX may help alleviate DLB-associated neurodegeneration.

The experimental work is sounding and the methodology used is appropriate.

Results are clear and figures are informative.

However all the figures should be redrawn using mean ± Standard deviation or even better using whiskers plot graphics.

The discussion is balanced and consistent with results.

References are appropriate.

English is fine.

Line 136: results must be reported as mean ± Standard deviation and not SEM

Line 159: the notes to figure 2 should be in one page.

Line 172: results must be reported as mean ± Standard deviation and not SEM

Line 212: results must be reported as mean ± Standard deviation and not SEM

Line 252: results must be reported as mean ± Standard deviation and not SEM

Line 432: results must be reported as mean ± Standard deviation and not SEM

Author Response

the manuscript ijms-3286572 entitled Protective Effects of Ambroxol on Aβ and α-Synuclein-Induced Neurotoxicity Through GCase Activation in HT-22 Hippocampal Neuronal Cells by Sheng‐Chieh Lin, aims to evaluate the Ambroxol (AMBX) neuroprotective effects in a cellular model of DLB, addressing the motivation to identify new therapeutic agents that target the underlying pathology of DLB. In this study, HT-22 cells were exposed to α-synuclein and Aβ, followed by AMBX treatment. The results showed that AMBX significantly improved cell viability and reduced apoptosis in cells co-treated with α-synuclein and Aβ. Additionally, AMBX restored GCase activity, promoting autophagy and reducing oxidative stress, which in turn mitigated α-synuclein aggregation and phosphorylation. These findings suggest that by activating GCase and enhancing autophagy, AMBX may help alleviate DLB-associated neurodegeneration.

The experimental work is sounding and the methodology used is appropriate.

Results are clear and figures are informative.

However all the figures should be redrawn using mean ± Standard deviation or even better using whiskers plot graphics.

The discussion is balanced and consistent with results.

References are appropriate.

English is fine.

Comment 1:

Line 136: results must be reported as mean ± Standard deviation and not SEM

Line 159: the notes to figure 2 should be in one page.

Line 172: results must be reported as mean ± Standard deviation and not SEM

Line 212: results must be reported as mean ± Standard deviation and not SEM

Line 252: results must be reported as mean ± Standard deviation and not SEM

Line 432: results must be reported as mean ± Standard deviation and not SEM

Response 1: We would like to express our sincere gratitude to the reviewer for their positive assessment of our work and for the valuable feedback provided. We are pleased that the reviewer found the experimental design and methodology sound, and that the results and figures were clear and informative. In response to the suggestion regarding figure presentation, we have made adjustments in the revised manuscript to improve clarity, including changes to display data as mean ± standard deviation (SD). We appreciate the recommendation to consider alternative formats, such as whisker plots, and will keep this in mind for future revisions to enhance data visualization. Once again, we thank the reviewer for their constructive comments and remain open to any further suggestions that could help us improve the quality of our work.

Reviewer 3 Report

Comments and Suggestions for Authors

The authors present an in vitro study using the immortalized mouse hippocampal cell line HT-22 transfected with plasmids expressing human amyloid beta (Aβ) and α-synuclein. The authors demonstrated that ambroxol, a mucolytic drug, ameliorates the deleterious effects of these two proteins involved in the pathology of neurodegenerative diseases through activation of glucocerebrosidase (GCase) with associated enhancement of autophagy.

The experiments are well planned and executed and the results are robust and consistent. In my opinion, the manuscript would be publishable with a few minor changes.

In general, most of the figures are extremely small and difficult to evaluate, even with 200% screen zoom enhancement.

I think that the abbreviations Aβ and GCase are not extended enough to be considered standard and therefore it might not be appropriate to use these abbreviations in the title. In addition, it sounds rare to abbreviate Aβ and not to abbreviate α-synunclein.

Figure 1B: I assume that pSwitch is the plasmid carrying the αsyn gene, but what is Mock, plasmid not carrying the gene encoding the protein?

Figure 1B, why does the image of the gel show up to 2.5 µM mifepristone and the graph bar up to 20 µM?

Figure 2B: It is not so obvious to me that AMBX significantly reduces the expression of apoptosis markers. Didn't you quantify the blots?

I missed a control in Figure 4 to test the effect of Conduritol B epoxide alone.

Please report the purity of ambroxol in section 4.

I missed a discussion of the potential translation of the reported findings to in vivo conditions. Does amboxol cross the blood-brain barrier? If not, what are the challenges to using this drug in vivo? Is 20 µM a physiologically feasible concentration? What is the concentration of ambroxol in the blood of patients using this mucolytic agent? Does ambroxol undergo biotransformation in the liver? If so, this could be an additional challenge.

Author Response

The authors present an in vitro study using the immortalized mouse hippocampal cell line HT-22 transfected with plasmids expressing human amyloid beta (Aβ) and α-synuclein. The authors demonstrated that ambroxol, a mucolytic drug, ameliorates the deleterious effects of these two proteins involved in the pathology of neurodegenerative diseases through activation of glucocerebrosidase (GCase) with associated enhancement of autophagy.

The experiments are well planned and executed and the results are robust and consistent. In my opinion, the manuscript would be publishable with a few minor changes.

Comment 1:

In general, most of the figures are extremely small and difficult to evaluate, even with 200% screen zoom enhancement.

Response 1: We would like to thank the reviewer for the valuable feedback regarding the figure sizes. In this revised version, we have reformatted the figures and enlarged certain images to improve readability. We hope these adjustments enhance the quality and ease of evaluation for the figures. We appreciate the reviewer’s suggestion to improve the presentation of our data.

Comment 2:

I think that the abbreviations Aβ and GCase are not extended enough to be considered standard and therefore it might not be appropriate to use these abbreviations in the title. In addition, it sounds rare to abbreviate Aβ and not to abbreviate α-synunclein.

Response 2: Thank you to the reviewer for pointing out the inconsistency with the abbreviations in the title. We have revised the title to remove abbreviations for clarity. Additionally, to ensure consistency throughout the manuscript, we have standardized the abbreviation of α-synuclein to “αSyn” in all instances. We appreciate the reviewer’s guidance in helping us improve the clarity and uniformity of our manuscript.

Comment 3:

Figure 1B: I assume that pSwitch is the plasmid carrying the αsyn gene, but what is Mock, plasmid not carrying the gene encoding the protein?

Response 3: Thank you for your question. In Figure 1B, both the control and experimental cells contain the pSwitch plasmid, which allows activation of gene expression upon mifepristone treatment. In the control group, the pGene vector is empty (lacking any gene insert), so no protein is expressed, serving as a baseline control. In the experimental group, the pGene vector carries the α-synuclein gene, enabling α-synuclein overexpression upon induction. We have clarified this in both the manuscript text and the figure legend to avoid any confusion. We appreciate the reviewer’s attention to detail, which has helped us improve our explanations.

Comment 4:

Figure 1B, why does the image of the gel show up to 2.5 µM mifepristone and the graph bar up to 20 µM?

Response 4: We apologize for the oversight and thank the reviewer for identifying this labeling error. We have corrected it in the revised figure. In Fig. 1B, the Western blot (WB) images show αSyn overexpression at 24 h and 48 h with different mifepristone concentrations (0, 0.5, 1, and 2.5 μM). It is evident that the induction at 48 h is more pronounced, so we used the 48 h induction time for subsequent experiments. The quantitative graph below corresponds to αSyn expression levels at 48 h, with mifepristone concentrations matching the WB data (0, 0.5, 1, and 2.5 μM).

Comment 5:

Figure 2B: It is not so obvious to me that AMBX significantly reduces the expression of apoptosis markers. Didn't you quantify the blots?

Response 5: We apologize for the oversight and thank the reviewer for pointing out this issue. The original labeling in Figure 2B was indeed incorrect, as the AMBX treatment sequence should have been indicated as "- - + -". The initial WB image was generated from an earlier set of data, where sample loading did not match the arrangement of other figures. To ensure consistency with the other figures, we have redone the Western blot analysis, and the corrected image is presented in this revised version. Additionally, as per the reviewer’s request, we have quantified the blots. The time taken to repeat this experiment was the primary reason for the delay in submitting this revised version. We deeply appreciate the reviewer’s attention to detail, which has helped us improve the accuracy and clarity of our results.

Comment 6:

I missed a control in Figure 4 to test the effect of Conduritol B epoxide alone.

Response 6: We apologize for the absence of a control experiment testing the effect of Conduritol B epoxide (CBE) alone in Figure 4. Due to time constraints, we were unable to conduct this additional experiment in time for the resubmission. However, we would like to reference the findings of Perez-Abshana et al. (Int J Mol Sci 2023; 24:10589), who demonstrated that treatment with CBE alone induces αSyn aggregation and an increase in pS129. Based on these findings, it is reasonable to infer that CBE treatment alone in αSyn-overexpressing neuronal cells would likely result in an increase in αSyn aggregation as well. We appreciate the reviewer’s suggestion, and we will consider incorporating this control in future studies to further validate our findings. Thank you for your understanding.

Comment 7:

Please report the purity of ambroxol in section 4.

Response 7: Thank you for the suggestion. According to the manufacturer’s specifications, the purity of Ambroxol used in our experiments is ≥ 98%. We have added this information in the "4. Materials and Methods" section of the revised manuscript. We appreciate the reviewer’s attention to detail, which has helped us enhance the clarity of our methodology.

Comment 8:

I missed a discussion of the potential translation of the reported findings to in vivo conditions. Does amboxol cross the blood-brain barrier? If not, what are the challenges to using this drug in vivo? Is 20 µM a physiologically feasible concentration? What is the concentration of ambroxol in the blood of patients using this mucolytic agent? Does ambroxol undergo biotransformation in the liver? If so, this could be an additional challenge.

Response 8: Thank you for this valuable comment regarding the translational implications of our findings.

  1. Blood-Brain Barrier (BBB) Permeability: Ambroxol is indeed capable of crossing the blood-brain barrier, making it a promising candidate for treating neurodegenerative conditions. This property has been demonstrated in clinical trials, where high-dose Ambroxol improved neurological outcomes in patients with neuronopathic Gaucher disease ​( Hollander et. al., Mol Genet Meta 2024;143:108556).
  2. Physiological Feasibility of 20 µM Concentration: While 20 µM was used in our in vitro experiments to assess efficacy, achieving this concentration in vivo can be challenging due to absorption, distribution, metabolism, and excretion processes. In clinical contexts, oral administration of Ambroxol as a mucolytic agent results in lower plasma concentrations, typically below 5 µM. Therefore, alternative dosing strategies or delivery methods might be required to achieve neuroprotective concentrations in the brain.
  3. Biotransformation in the Liver: Ambroxol undergoes hepatic metabolism primarily via glucuronidation, which can reduce the amount of active drug reaching the target tissues. Indeed, this biotransformation may pose a challenge for achieving sustained therapeutic levels in the brain, particularly at higher doses required for neuroprotection.

We have expanded the discussion section to address these points, considering the translational challenges and feasibility of Ambroxol as a neuroprotective agent in vivo. We appreciate the reviewer’s insightful suggestions, which have strengthened the contextual relevance of our study.

Round 2

Reviewer 1 Report

Comments and Suggestions for Authors

I don't have any concerns.